# OMNICUSTOM: A MULTIMODAL-DRIVEN ARCHITECTURE FOR CUSTOMIZED VIDEO GENERATION

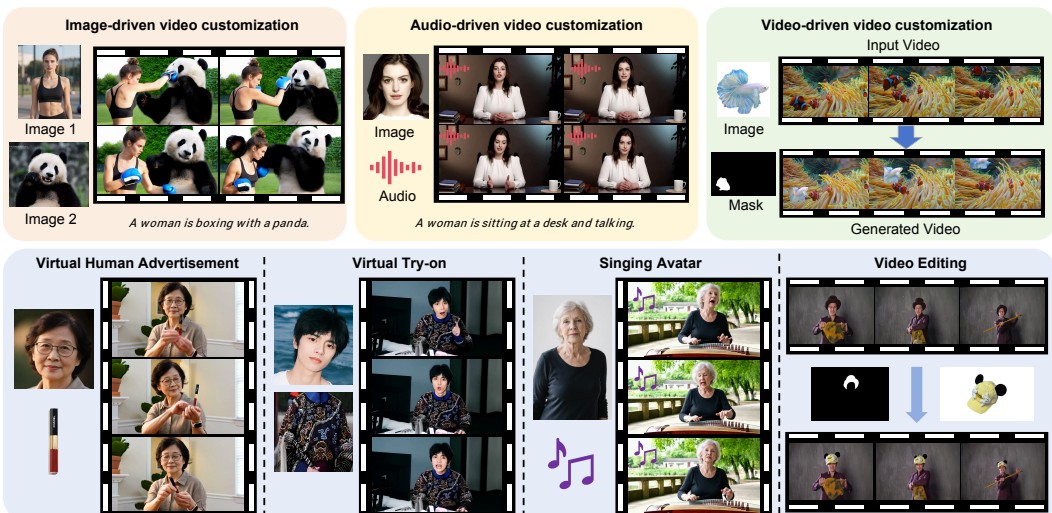

Figure 1: OmniCustom facilitates multi-modal driven video customization, allowing for the generation of videos based on text, images, audio, and video inputs. It supports a wide range of applications, such as virtual human advertisements, virtual try-ons, singing avatars, and video editing, significantly enhancing the controllability of subject-centric video generation.

## ABSTRACT

Customized video generation aims to produce videos featuring specific subjects under flexible user-defined conditions, yet existing methods often struggle with identity consistency and limited input modalities. In this paper, we propose Omni-Custom, a multi-modal customized video generation model that emphasizes subject consistency while supporting image, audio, video, and text conditions. Built upon HunyuanVideo, OmniCustom introduces an identity-enhanced text-image conditioning module based on LLaVA for improved multi-modal understanding, and an image ID enhancement module that leverages temporal concatenation to reinforce identity features. To enable flexible audio- and video-driven customization, we further propose modality-specific injection modules. Our identity-disentangled AudioNet injects temporally aligned audio features into video latents via spatial cross-attention, enabling precise audio control. For video-driven generation, we design an identity-disentangled video injection module that projects conditional video into the latent space and efficiently aligns video features with latents for seamless integration. Extensive experiments on single- and multi-subject scenarios show that OmniCustom significantly outperforms state-of-the-art methods in ID consistency, realism, and text-video alignment. We further demonstrate its robustness on downstream tasks such as audio- and video-driven customized video generation, highlighting the effectiveness of our multi-modal conditioning and identity-preserving strategies for customized video generation.

## 1 INTRODUCTION

The field of video generation has undergone rapid advancement in recent years, driven by the proliferation of both open-source Xu et al. (2025); Hu et al. (2024); Zhou et al. (2024b); Xue et al.

(2025); Huang et al. (2024b) and commercial video-generation models Vidu (2025); Keling (2025); Pika (2025); Hailuo (2025). These advancements have significant real-world implications, ranging from content creation in the entertainment industry to applications in education, advertising, and more. However, a critical limitation persists: the lack of precise controllability in current models. Generating videos that adhere to users' specific requirements is still challenging, which restricts their potential applications in real-world scenarios where fine-grained customization is essential.

Customized video generation focuses on creating videos with specific subjects. Existing methods like ConsisID (Yuan et al., 2024) and MovieGen (Polyak et al., 2025) generate videos for a single human ID but cannot handle arbitrary objects. Other approaches, such as ConceptMaster (Huang et al., 2025), Video Alchemist (Chen et al., 2025a), Phantom (Liu et al., 2025), and SkyReels-A2 (Fei et al., 2025), extend to multi-subject generation but struggle with maintaining subject consistency and video quality, and are limited by single-modality (image-driven) inputs. Recently, VACE (Jiang et al., 2025), based on the *Wan* model (Wang et al., 2025), introduced a multi-modal-conditioned framework, but its extensive training tasks affect ID consistency.

To address these limitations, we propose OmniCustom, a multi-modal video customization model built upon HunyuanVideo Kong et al. (2024), which generates customized videos with high subject consistency while supporting diverse multi-modal inputs, including image identities, audio conditions, video backgrounds, and text prompts. We first introduce a novel **identity-enhanced text-image condition module** that combines the multi-modal understanding capabilities of LLaVA with the temporal modeling strengths of a pretrained video generation model. By establishing strong correlations between text and image identities and reinforcing identity information during generation, this module significantly improves subject consistency. Building on this foundation, OmniCustom further supports both **audio-driven** and **video-driven** customized video generation. To enable audio and video injection without affecting image-based conditioning, we design an **identity-disentangled AudioNet**, which first aligns audio features with video latents along the temporal axis, and then injects audio conditions frame by frame using spatial cross-attention, effectively incorporating audio information. Additionally, we propose an **identity-disentangled video injection module,** which projects the input video into the same space as the noisy video latents. Our module then employs an efficient feature-alignment addition operation to enable video conditioning without introducing extra computational overhead. Since the audio and video injection modules are isolated from the image injection module, they avoid mutual interference, allowing OmniCustom to flexibly generate high-quality, subject-consistent videos under diverse multi-modal conditions.

OmniCustom has been rigorously evaluated on single-subject and multi-subject consistency generation. We compare it with existing open-source and closed-source methods, conducting comprehensive comparisons across key metrics such as ID consistency, generation quality, and video-text alignment. The experimental results show that OmniCustom outperforms all existing methods in customized video generation. In addition, we validate its robustness through extensive experiments on audio and video-driven video customization, highlighting the superior performance of our method. Thanks to its strong identity preservation and multi-modal control capabilities, OmniCustom shows great potential for real-world applications such as virtual human advertising, virtual try-on, and fine-grained video editing. These results demonstrate the effectiveness of our OmniCustom, providing a solid foundation for future research in controllable subject-consistent video generation. The main contributions can be summarized as four-fold:

- We propose **OmniCustom**, a novel **multi-modal customized video generation model** that robustly maintains subject consistency. OmniCustom supports diverse conditioning inputs, including image identities, audio, video backgrounds, and text prompts, enabling flexible and subject-consistent video generation across various modalities.

- We design an **identity-enhanced text-image condition module** that integrates the multi-modal understanding capabilities of LLaVA with the temporal modeling strengths of the pretrained video generation model. By establishing strong correlations between text and image identities and reinforcing identity information during generation, this module significantly improves subject consistency.

- We propose identity-disentangled **audio and video condition injection module** to enable both audio-driven and video-driven customized video generation. Specifically, an AudioNet employs spatial cross-attention for efficient audio injection, while a video alignment-based

injection mechanism is utilized for effective video conditioning. Together, these approaches achieve flexible and effective multi-modal video customization.

- Extensive experiments show that OmniCustom achieves state-of-the-art customized video generation ability, and remains robust and versatile in audio- and video-driven customization for real-world applications such as virtual human advertising and video editing.

## 2 RELATED WORK

### 2.1 VIDEO GENERATION MODEL

Recent advancements in video generation have been significantly driven by diffusion models, which have successfully evolved from static image synthesis (Rombach et al., 2022; Li et al., 2024c; Labs, 2024; Hu et al.) to dynamic spatio-temporal modeling (Hong et al., 2022; Zhang et al., 2023b). The field has witnessed substantial progress with large-scale frameworks (Liu et al., 2024; Yang et al., 2024; Kong et al., 2024; Wang et al., 2025; Zhou et al., 2024a), which demonstrate unprecedented high-quality content creation and a diverse array of generated results through extensive training on video-text pairs. However, existing methods primarily concentrate on either text-guided video generation (Lin et al., 2025) or video generation based on a single reference image (Gao et al., 2023; Xu et al., 2025). These approaches often struggle to provide fine-grained control over the generated content and precise concept-driven editing. This limitation continues to exist despite advancements in multi-condition control. While pioneering work such as VACE (Jiang et al., 2025) enables multi-condition capabilities through multi-modal modeling, it fails to maintain identity consistency due to the excessive number of training tasks. In this study, we meticulously design a multi-condition-driven model that incorporates various modalities, including images, videos, audios, and texts, while also emphasizing subject-consistency generation.

### 2.2 VIDEO CUSTOMIZATION

Video customization aims at generating videos containing the given subjects. Early methods (Chefer et al., 2024; Wu et al., 2025; Wang et al., 2024b; Chen et al., 2024) follow image customization methods like DreamBooth (Ruiz et al., 2023), where they embed the subject image into the textual space, and fine-tune the pretrained text-to-video model to generate the corresponding subjects. However, these methods need to train one model for one subject image, which pose a challenge in real-time and large-scale applications. Therefore, recent works aim at training an end-to-end customization model. IDanimator, ConsisID, and Moviegen first propose to generate ID-consistent videos for humans. VideoBooth (Jiang et al., 2024c) further extends to generate subjects for arbitrary subjects. To enable multi-subject customization, some works, including ConceptMaster (Huang et al., 2025), Video Alchemist (Chen et al., 2025a), Phantom (Liu et al., 2025), SkyReels-A2 (Fei et al., 2025), VACE (Jiang et al., 2025), and PolyVivid Hu et al. (2025) take several subject images as input, and then generate videos including the given multiple subjects. However, the complex interactions among multiple subjects pose significant challenges to maintaining subject consistency. Additionally, these methods focus solely on generating videos from single-modal inputs, specifically subject images. In contrast, our approach enables customized video generation with multi-modal conditional inputs (text, image, audio, and video) while maintaining a high degree of consistency, resulting in superior performance and broader applicability.

## 3 VIDEO CUSTOMIZATION

**OmniCustom is a multi-modal customized generation model** centered on subject consistency. It enables the generation of subject-consistent videos conditioned on text, images, audio, and video inputs, as shown in Fig. 2. Specifically, OmniCustom introduces an identity-enhanced text-image condition module which employ LLaVA to facilitate interaction between images and text, allowing identity information from images to be effectively integrated into textual descriptions. Additionally, an identity enhancement module is proposed, which concatenates image information along the temporal axis and leverages the video model's efficient temporal modeling ability to enhance subject identity throughout the video. To support conditional injection of audio and video, OmniCustom designs distinct injection mechanisms for each modality, which are effectively disentangled with the

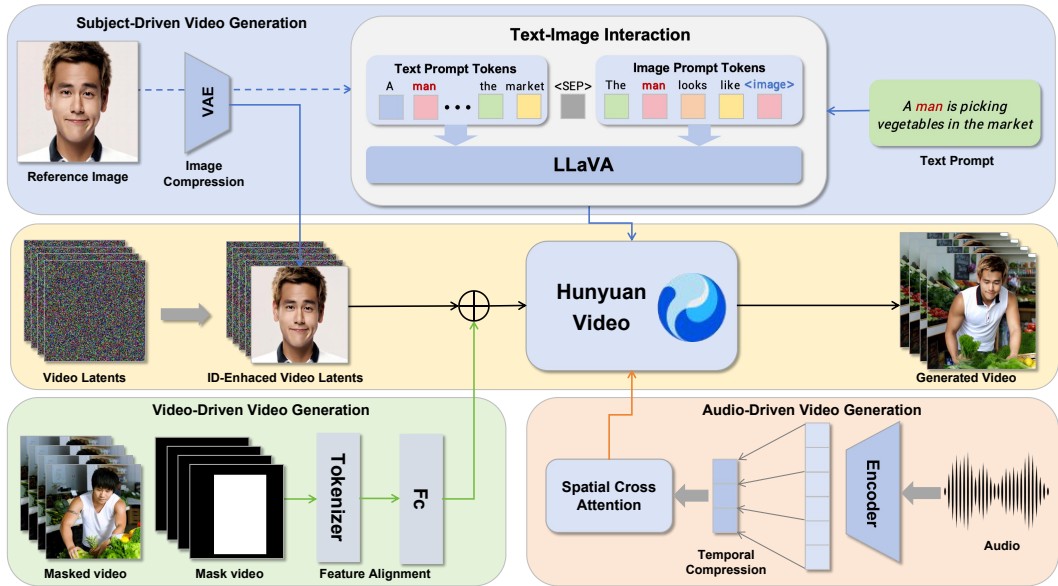

Figure 2: The main framework of OmniCustom, where we can generate identity-consistent videos conditioned on text, image, audio, and video.

image-level identity condition module. OmniCustom ultimately achieves decoupled control over image, audio, and video conditions, demonstrating great potential in subject-centric multi-modal video customization.

## 3.1 MULTI-MODAL DATA CONSTRUCTION

To enable multi-modal customized video generation, we first construct a multi-modal video customization dataset. We collect video data from both open-source datasets (Wang et al., 2024a) and our private dataset. We then filter out videos containing subtitles, watermarks, and logos. Subsequently, we use the koala-36M model (Wang et al., 2024a) to eliminate low-quality videos. To build the customization dataset, we employ QwenVL (Bai et al., 2025b) to identify subjects in the videos based on captions. We then use GroundingDINO (Caron et al., 2021) and SegmentAnything 2 (Ravi et al., 2024) to segment the corresponding subject images. For audio data, we first utilize LatentSync (Li et al., 2024a) to find videos with high audio-video synchronization and then employ Whisper (Radford et al., 2023) to extract audio features. More details are provided in the supplementary material.

## 3.2 IMAGE-DRIVEN VIDEO CUSTOMIZATION

At the core of OmniCustom is the task of generating videos conditioned on an input image $I$ representing a specific identity and a textual description $T$. A key challenge is enabling the model to effectively comprehend the identity information embedded in the image and integrate it with the textual context for interactive understanding. To address this, OmniCustom introduces an identity-enhanced text-image conditioning module, which consists of a LLaVA-based text-image interaction component and an identity enhancement module. This module facilitates joint modeling of visual and textual inputs while enhancing the consistency of subject identities.

**LLaVA-based text-image interaction.** In the context of video customization, effectively integrating image and text information has been a key challenge for previous customization methods (Fei et al., 2025; Jiang et al., 2025; Liu et al., 2025), which typically treat image and text as two independent modalities. By leveraging the text comprehension capabilities of LLaVA (Liu et al., 2023a), as adopted in HunyuanVideo (Kong et al., 2024), we extend the original text input of HunyuanVideo to incorporate both text and images. This enables effective image-text interaction and understanding, capitalizing on LLaVA's strong multimodal comprehension abilities.

Specifically, given a text input $T$ and an image input $I$ with a corresponding description word $T_I$ in the text, we design a template to facilitate interaction between the text and image. We explore two types of templates: (1) the *image-embedded template*, where the description word $T_I$ in the text is replaced with the image token `<image>` (e.g., for the text prompt "A man sitting on the grass," if the identity image corresponds to "man," the resulting template is "A `<image>` sitting on the grass"); and (2) the *image-appended template*, where the image token is placed after the text prompt by adding an identity prompt, "The $T_I$ looks like `<image>`" (for the above example, the resulting template is "A man sitting on the grass. The man looks like `<image>`"). After processing, the image token `<image>` is replaced by $24 \times 24$ image hidden features extracted by LLaVA's image encoder. Since the image tokens are significantly longer than the text tokens, to prevent the image features from overwhelming text comprehension, we insert a special token `<SEP>` between the text prompt and the image prompt. This helps the LLaVA model retain the information from the text prompt while establishing a connection between the textual description and the image identity.

**Identity Enhancement.** The LLaVA model, as a multi-modal understanding framework, is designed to capture the correlation between text and image, primarily extracting high-level semantic information such as category, color, and shape, while often overlooking finer details. However, in video customization, identity is significantly determined by these image details, making the LLaVA branch alone insufficient for identity preservation. To address this, we propose an identity enhancement module. By concatenating video latents with the target image over the time axis and leveraging the video model's efficient information transmission capability in the temporal dimension, we can effectively enhance video identity consistency.

Specifically, we first resize the image to match the video frame size. We then employ the pretrained causal 3D-VAE from HunyuanVideo to map the image $I$ from image space to latent space. With the image latent $z_I \in \mathbb{R}^{wh \times c}$, where $wh$ represents the width and height of the latent and $c$ is the feature dimension, we concatenate the noisy video latent $z_t \in \mathbb{R}^{fwh \times c}$ (where $f$ is the number of video frames) and the image latent $z_I$ along the first sequence dimension to obtain a new latent $z = \{z_I, z_t\} \in \mathbb{R}^{(f+1)wh \times c}$. Given the pretrained Hunyuanvideo's strong prior in modeling temporal information, identity can be efficiently propagated along the time axis. Consequently, we assign the concatenated image latent with a 3D-RoPE (Su et al., 2024) along the time series. In the original Hunyuan video, the video latent is assigned a 3D-RoPE along the time, width, and height axes; for a pixel located at $(f, i, j)$ (where $f$ is the frame index, $i$ is the width, and $j$ is the height), it receives a RoPE with $RoPE(f, i, j)$. For the image latent, to enable effective identity broadcasting along the time series, we position the image latent at the $-1$-th frame, preceding the first frame with time index $0$. Furthermore, inspired by Omnicontrol (Tan et al., 2024) in controllable image generation, to prevent the model from simply copying and pasting the target image into the generated frames, we introduce a spatial shift for the image latents, where:

$$RoPE_{z_I}(f, i, j) = RoPE(-1, i + w, j + h). \tag{1}$$

**Multi-subject Customization.** For multi-subject customization, we utilize the trained single-subject customization model as a foundation and subsequently fine-tune it to accommodate the multi-subject customization task. Specifically, we have several condition images $\{I_1, I_2, \ldots, I_m\}$, each with corresponding text descriptions $\{T_{I,1}, T_{I,2}, \ldots, T_{I,m}\}$. For each image, we template them as "the $T_{I,k}$ looks like `<image>`" and model the text-image correlation using the LLaVA model. Additionally, to enhance image identity, we encode all images into latent space to obtain image latents $\{z_{I,1}, z_{I,2}, \ldots, z_{I,m}\}$ using 3D-VAE, and then concatenate them with the video latent. To differentiate between various identity images, we assign the $k$-th image a time index of $-k$, which is associated with a 3D-RoPE:

$$RoPE_{z_{I,k}}(f, i, j) = RoPE(-k, i + w, j + h). \tag{2}$$

### 3.3 Audio-driven video customization

**Audio-driven video customization.** Audio is an indispensable component in video generation, with extensive research dedicated to using audio as a condition to drive video creation. Among these, *audio-driven human animation* represents an important research topic. Existing models (Jiang et al., 2024a; Ji et al., 2024) for audio-driven human animation typically use a human image and audio as input to animate the character in the image to speak the corresponding speech. However, this image-to-video paradigm results in generated videos where the character's posture, attire, and setting

remain consistent with the input image, limiting the ability to generate videos of the target character in different postures, attire, and settings. This limitation restricts their application. Leveraging OmniCustom's effective capture and maintenance of character identity information, we further integrate audio input to enable the generation of videos where the character speaks the corresponding audio in a text-described scene, allowing for more flexible and controllable speech-driven virtual human generation, which we call *audio-driven video customization*.

**Identity-disentangled AudioNet.** To effectively decouple audio signals from identity-related information, we propose the **identity-disentangled AudioNet**. As outlined in Section 3.2, identity cues are primarily introduced through the text modality via LLaVA and further reinforced by token concatenation along the latent temporal dimension. To mitigate potential interference between the audio and identity modalities, AudioNet adopts an alternative conditioning strategy explicitly designed to prevent entanglement with identity information. Given an audio-video sequence consisting of $f'$ frames, we extract audio features for each frame, yielding a feature tensor of shape $f' \times 4 \times c$, where 4 denotes the number of tokens per audio frame. The corresponding video latent representations are temporally compressed by a pretrained 3D VAE into $f$ frames, with $f = \left\lfloor \frac{f'}{4} \right\rfloor + 1$—where the additional 1 accounts for the initial, uncompressed frame, and 4 is the temporal compression ratio. Furthermore, to incorporate identity information, an identity image is concatenated at the beginning, resulting in a video latent of $f + 1$ frames. To ensure temporal alignment between the audio features and the compressed video latent, we first pad the audio feature sequence prior to the initial frame, producing a total of $(f + 1) \times 4$ audio frames. We then aggregate every four consecutive audio frames into one, resulting in a temporally aligned audio feature tensor $f_A$ that matches the structure of the video latent representation.

$$f_A = Rearrange(f_{A,0}) : [b, (f+1) \times 4, 4, c] \rightarrow [b, (f+1), 16, c]. \tag{3}$$

With the temporally aligned audio features $f_A$, we introduce audio information into the video latent representation $z_t$ using a cross-attention mechanism. To prevent interference across different time steps, we adopt a **spatial cross-attention** strategy that performs audio injection separately for each time step. Specifically, each audio frame interacts only with the spatial tokens of its temporally aligned video frame, and cross-attention is applied independently at each temporal index. To this end, we decouple the temporal dimension from the spatial dimensions of the video latent and apply attention solely along the spatial axes:

$$\begin{aligned} z'_{t,A} = Rearrange(z_t) : &[b, (f+1)wh, c] \rightarrow [b, f+1, wh, c], \\ z''_{t,A} = z'_{t,A} + &\lambda_A \times CrossAttn(f_A, z'_t), \\ z_{t,A} = Rearrange(z''_{t,A}) : &[b, f+1, wh, c] \rightarrow [b, (f+1)wh, c], \end{aligned} \tag{4}$$

where $\lambda_A$ is a weight to control the influence of the audio feature.

## 3.4 VIDEO-DRIVEN VIDEO CUSTOMIZATION

In practical video creation, editing is a fundamental task that often involves modifying subject appearance and motion, which naturally aligns with OmniCustom's subject-level generation capabilities such as replacement and insertion. However, videos contain rich spatiotemporal information, making both content extraction and integration challenging. Existing methods like VACE (Jiang et al., 2025) rely on adapter-based conditioning, resulting in doubled computational cost. Other approaches (Bai et al., 2025a) concatenate conditioning and generated video latents along the temporal axis, causing quadratic growth in attention computation. To address these limitations, OmniCustom adopts an efficient video condition injection strategy that decouples video information from image and audio modalities. Specifically, it compresses the conditioning video using a pretrained causal 3D-VAE, aligns the resulting features with noisy video latents via feature alignment, and directly adds the aligned features to the latent representation. This approach enables effective conditioning with minimal computational overhead.

**Video-Latent Feature Alignment.** The conditioning video serves as a clean, noise-free input, whereas the video latents are obtained from a noisy encoding process. To improve video condition injection, we first perform feature alignment between the conditioning video and the video latents. Specifically, the conditioning video is encoded using the pretrained causal 3D-VAE encoder, followed by compression and serialization via the pretrained video tokenizer in HunyuanVideo. Then, a fully

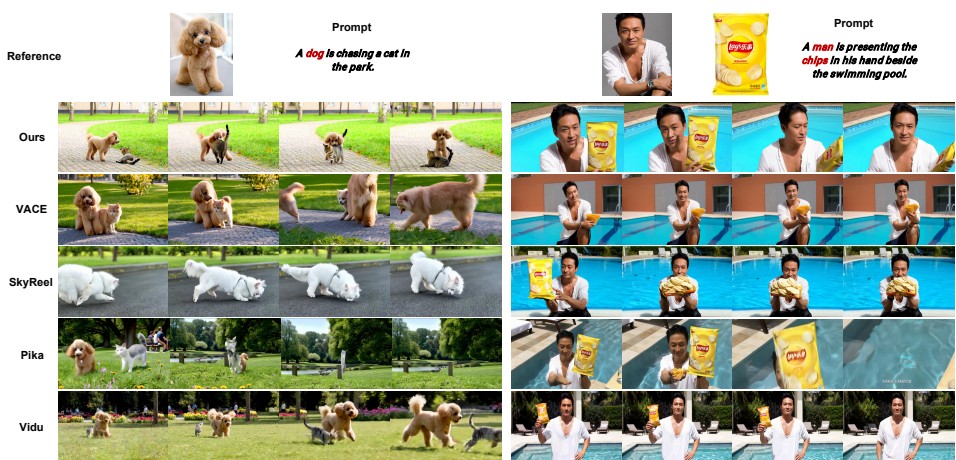

Figure 3: Comparison of single-subject and multi-subject video customization.

connected network maps the conditioning video features into the latent space, achieving feature alignment with the video latents.

**Identity-Disentangled Video Conditioning.** Building upon the prior feature alignment, we introduce an identity-disentangled conditioning mechanism that injects video information independently from image and audio modalities. Specifically, the aligned conditioning video features are directly added to the video latents on a frame-by-frame basis along the temporal dimension, preserving the original feature dimensions. This approach maintains the same latent shape as the original video latents and incurs no additional computational overhead during inference. Experiments show that this addition-based method effectively fuses video features, enabling efficient and high-fidelity video condition incorporation.

## 4 EXPERIMENT

### 4.1 COMPARISON ON SINGLE-SUBJECT VIDEO CUSTOMIZATION

**Baselines.** We compare OmniCustom with the state-of-the-art video customization methods, including commercial products (Vidu 2.0 (Vidu, 2025), Keling 1.6 (Keling, 2025), Pika (Pika, 2025), and Hailuo (Hailuo, 2025)) and open-sourced methods (Skyreels-A2 (Fei et al., 2025) and VACE (Jiang et al., 2025)). More implementation details are presented in #Suppl.

**Single-subject customization comparisons.** We present a comparison of state-of-the-art methods in Fig. 3 (left) for single-subject video customization. It is evident that VACE, SkyReels-A2, and Pika exhibit poor identity preservation, with SkyReels-A2 failing to generate the specified dog in the video. Additionally, VACE sometimes struggles to distinguish subjects, merging the generated cat and dog into a single indistinct entity. Vidu shows commendable generation quality and identity consistency among the methods compared, yet there remains room for improvement in identity consistency. In contrast, OmniCustom excels in generating videos with high identity consistency while maintaining superior generation quality and diversity, highlighting its advantage over other methods.

**Multi-subject customization comparisons.** We also compare the multi-subject customization ability with the existing methods. The comparative results are presented in Fig. 3 (right). Pika can generate the specified subjects but exhibits instability in video frames, with instances of a man disappearing in one scenario and a woman failing to open a door as prompted. Vidu and VACE partially capture human identity but lose significant details of non-human objects, indicating a limitation in representing non-human subjects. SkyReels-A2 experiences severe frame instability, with noticeable changes in chips and numerous artifacts in the right scenario. In contrast, our OmniCustom effectively captures both human and non-human subject identities, generates videos that adhere to the given prompts, and maintains high visual quality and stability.

**Quantitative comparison.** We conduct a quantitative comparison between the state-of-the-art methods in Tab. 1. Our OmniCustom achieves the best ID consistency and subject consistency. It also achieves comparable results in prompt following and generation quality. Hailuo (Hailuo, 2025) has

Table 1: The quantitative comparison results on single-subject video customization. **Bold** and underline represent optimal and sub-optimal results, respectively.

| Models | Face-Sim ↑ | DINO-Sim ↑ | CLIP-B-T ↑ | CLIP-L-T ↑ | FVD ↓ | Temp-Consis ↑ |
|---|---|---|---|---|---|---|
| VACE 1.3B (Jiang et al., 2025) | 0.204 | 0.569 | 0.308 | 0.260 | 1490 | **0.967** |
| Skyreels-A2 (Fei et al., 2025) | 0.402 | 0.579 | 0.295 | 0.256 | 1904 | 0.942 |
| Pika (Pika, 2025) | 0.363 | 0.485 | 0.305 | 0.255 | **1247** | 0.928 |
| Vidu 2.0 (Vidu, 2025) | 0.424 | 0.537 | 0.300 | 0.250 | 1698 | 0.961 |
| Keling1.6 (Keling, 2025) | 0.505 | 0.580 | 0.285 | 0.239 | 1348 | 0.914 |
| Hailuo (Hailuo, 2025) | 0.526 | 0.433 | **0.314** | **0.266** | 1485 | 0.937 |
| **OmniCustom (Ours)** | **0.624** | **0.607** | 0.309 | 0.261 | 1305 | 0.961 |

Figure 4: Results on audio and video-driven video customization (Refer to #Suppl for more results).

the best clip score because it can follow text instructions well with only ID consistency, sacrificing the consistency of full-body photos (the worst DINO-Sim). In terms of FVD and temporal consistency, our model achieves competitive scores, indicating its good generation quality.

## 4.2 EXPERIMENTS ON MULTI-MODAL VIDEO CUSTOMIZATION

**Audio-driven video customization.** Previous audio-driven human animation methods input a human image and an audio, where the human posture, attire, and environment remain consistent with the given image and cannot generate videos in other gestures and environments, which may restrict their application. In comparison, our OmniCustom enables audio-driven human customization, where the character speaks the corresponding audio in a text-described scene and posture, allowing for more flexible and controllable audio-driven human animation. The generated results are illustrated in Fig. 4. OmniCustom produces videos that closely align with the given prompts while preserving character identities. It demonstrates effective audio alignment and prompt following, which can significantly enhance its application in live streaming and advertising. Additionally, it can generate videos featuring diverse scenes and postures, such as the example in the Ming Dynasty (row 2), where characters are automatically dressed in period-appropriate attire without explicit prompts. This demonstrates OmniCustom's robust world modeling and generalization capabilities. In summary, our audio-driven OmniCustom can generate videos across various scenes and postures specified by text prompts with high diversity, while keeping the identity well.

**Video-driven video customization.** Leveraging its strong subject consistency, OmniCustom also supports video-driven video customization, which can be regarded as subject-centric video editing, enabling a broad range of application scenarios. The results are presented in Fig. 4, where a source video, object masks indicating regions to be replaced, and a target subject image are provided as inputs. It can be seen that our OmniCustom can replace the target subjects well, with high subject consistency with the given subject image, and good interaction with the video background, demonstrating its superior performance in video editing tasks. More comparisons are shown in the #Suppl.

**Quantitative comparisons on multi-modal video customization.** As our model is the first one to incorporate audio and video in video customization task, there are currently no existing methods or benchmarks for direct quantitative comparison in these scenarios. Nevertheless, to provide a more systematic evaluation, we have computed standalone quantitative metrics for our approach in both audio-driven and video-driven customization tasks in Tab. 2. We observe that the audio-driven

Table 2: Comparison of our model's performance across multi-modal video customization tasks.

| Models | Face-Sim ↑ | DINO-Sim ↑ | CLIP-B-T ↑ | CLIP-L-T ↑ | FVD ↓ | Temp-Consis ↑ |
|---|---|---|---|---|---|---|
| OmniCustom (Image only) | 0.624 | 0.607 | 0.309 | 0.261 | 1305 | 0.961 |
| OmniCustom (Audio-driven) | 0.629 | 0.615 | 0.294 | 0.241 | 1457 | 0.951 |
| OmniCustom (Video-driven) | - | 0.596 | 0.268 | 0.241 | 968 | 0.854 |

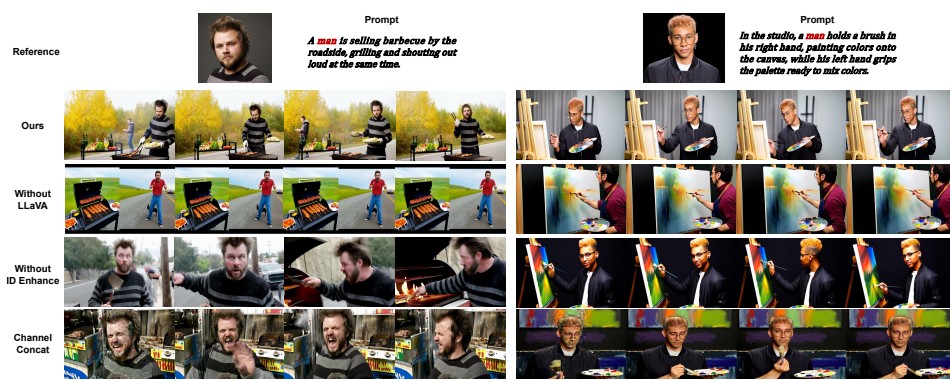

Figure 5: Ablation study on the proposed modules in OmniCustom.

customization achieves even better identity preservation metrics (e.g., Face-Sim and DINO-Sim) compared to the image-only setting, indicating the effectiveness of our approach in decoupling audio and identity information. For the video-driven setting, the CLIP scores for video-driven customization are slightly lower, as most of the video content is directly provided by the conditioning video, which may not always align perfectly with the target prompt. Moreover, since the conditioning video is real, the video-driven setting achieves a better FVD score, reflecting higher overall video quality.

## 4.3 ABLATION STUDY

We conduct ablation studies on subject customization, where we compare with three ablated models: (1) the model without LLaVA; (2) the model without identity enhancement; (3) the model with identity enhancement by channel-level concatenation. The results, presented in Fig. 5, reveal that the model without LLaVA exhibits poor identity preservation, indicating that LLaVA not only conveys prompt information but also extracts key identity features. The model without LLaVA fails to capture any significant details from the target image. Additionally, the model with LLaVA but lacking identity enhancement captures global identity information but misses detailed identity features, demonstrating the effectiveness of the identity enhancement module in refining identity details. Finally, the model using channel concatenation instead of temporal concatenation shows poor generation quality. Although it captures identity well, it suffers from a severe blurring effect in the initial frames, similar to results from Vidu (Vidu, 2025). This suggests that temporal concatenation aids in effectively capturing target information through strong temporal modeling priors and minimizes the impact on generation quality. In summary, our model successfully captures both global and local identity details while ensuring high generation quality, underscoring the effectiveness of our design.

## 5 CONCLUSION

In this paper, we propose OmniCustom, a multi-modal customized video generation model that addresses the key challenges of subject consistency and flexible controllability in video generation. By integrating an identity-enhanced text-image condition module and novel identity-disentangled audio and video injection modules, OmniCustom enables high-quality, subject-consistent video generation under diverse user-defined conditions, including image, audio, video, and text inputs. Extensive experiments on both single- and multi-subject scenarios demonstrate that OmniCustom outperforms existing methods in ID consistency, generation quality, and text-video alignment. Our model also shows strong robustness in audio- and video-driven customization tasks, confirming its versatility and practical value. These results highlight the effectiveness of our multi-modal conditioning and identity-preserving strategies for controllable, subject-consistent video generation.

ETHICS STATEMENT

In accordance with the ICLR Code of Ethics, we have carefully considered the societal impact of our work on OmniCustom. Our primary goal is to advance the field of controllable video generation for beneficial applications, such as enhancing creative expression, improving educational tools, and creating personalized content. We believe this technology holds significant potential for positive contributions across various industries.

We acknowledge our professional responsibility to address the potential for misuse. To uphold the principles of fairness, non-discrimination, and respect for individual rights, our research was conducted using publicly available datasets or datasets with appropriate licenses, ensuring that we did not use private or sensitive personal data without consent. We recognize that large-scale datasets may contain inherent societal biases, which our model could inadvertently learn. We are committed to transparency about this limitation and encourage future research focused on identifying and mitigating such biases to ensure equitable performance across different demographic groups.

To promote the responsible application of our work, we plan to release our code and models under a responsible AI license that explicitly prohibits use for malicious purposes, such as creating non-consensual content, spreading misinformation, or engaging in harassment. By making our methods public, we also aim to contribute positively to the research ecosystem, enabling the community to develop more effective detection and content provenance techniques. We believe that fostering an open and collaborative research environment is essential for developing shared norms and technical safeguards that guide the deployment of generative technologies for the benefit of society.

REPRODUCIBILITY STATEMENT

To ensure the transparency and reproducibility of our research, we provide a detailed account of our methodology, experimental setup, and resources. We are committed to making our work accessible to the research community to facilitate verification and future advancements.

- **Code and Models:** We will release the source code for OmniCustom, including training and inference scripts, upon the publication of this paper. The code will be made available in a public GitHub repository under an open-source license. We also plan to release the pretrained model weights for our proposed modules, including the identity-enhanced text-image condition module, the AudioNet, and the video injection module, to allow for direct replication of our results.

- **Datasets:** For the training and evaluation data, we were careful to only include videos that—to the best of our knowledge—were intended for free use and redistribution by their respective authors. That said, we are committed to protecting the privacy of individuals who do not wish their videos to be included in our datasets and will honor removal requests.

- **Implementation Details:** All critical implementation details and hyperparameters are provided in the main paper and the appendix. This includes the architecture of our proposed modules, optimizer settings, learning rates, batch sizes, training iterations, and so on. The appendix will contain a comprehensive illustration of all hyperparameters required to reproduce our key experiments.

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

# A APPENDIX

## A.1 OVERVIEW

In this supplementary material, we present additional implementation details, extended experimental results, and further analyses, organized as follows:

- Implementation details (Sec. A.2);
- Multi-modal data construction (Sec. A.3);
- More single-subject comparison results (Sec. A.4);
- Quantitative multi-subject comparison results (Sec. A.5);
- More results on audio-driven video customization (Sec. A.6).
- More results on video-driven video customization (Sec. A.7);
- Robustness to Diverse Real-World Scenarios (Sec. A.8);
- More Applications (Sec. A.9);
- Limitations and societal impacts (Sec. A.10).

We have also included a project page in the supplementary materials; please refer to it for additional video results. Note that, due to the 100MB size limit for supplementary files, the videos in the project page have been downsampled to a lower resolution. Our default resolution is 720p. Moreover, the code is also provided in the supplementary material.

## A.2 IMPLEMENTATION DETAILS

**Training Loss.** In the training process, we adopt the Flow Matching (Lipman et al., 2022) framework to train the video generation models. For training, we first acquire the video latent representation $z_1$ and the corresponding identity image $I$. Then, we sample $t \in [0, 1]$ from a logit-normal distribution (Esser et al., 2024) and initialize the noise $z_0 \sim N(0, I)$ according to the Gaussian distribution. After that, we construct the training sample $z_t$ through linear interpolation. The model aims to predict the velocity $u_t = \frac{dz_t}{dt}$ conditioned on the target image $I$, which is used to guide the sample $z_t$ towards $z_1$. The model parameters are optimized by minimizing the mean-squared error between the predicted velocity $v_t$ and the real velocity $u_t$, and the loss function is defined as:

$$\mathcal{L}_{generation} = \mathbb{E}_{t, x_0, x_1} \| v_t - u_t \|^2 . \tag{5}$$

**Training details.** OmniCustom is a multi-modal customized video generation model focused on subject consistency. To this end, we first train OmniCustom on the image-driven customization task with a 720P-resolution, which consists of two stages: we begin with 10,000 iterations of single-subject customization training, followed by 5,000 iterations of multi-subject customization training. Specifically, due to the large latent space of 720P videos, we initially train the model on 540P videos for the first 2,000 single-subject iterations to facilitate faster convergence. The model is trained with the AdamW optimizer with a learning rate $1e-5$. Furthermore, to enhance the model's representational capacity and enable it to capture a broader range of complex patterns, we fully fine-tune the weights of the pretrained HunyuanVideo model (Kong et al., 2024). This approach allows us to unlock the model's full potential and achieve superior video customization results. Once the image-driven customization task is well-trained, we proceed to train the model on both audio-driven and video-driven video customization tasks for an additional 5,000 iterations, which equips the model with multi-modal conditioning capabilities while preserving its original image-driven performance. All training is conducted on 256 GPUs with at least 80GB memory each, using a batch size of 64. Each training sample is distributed across 4 GPUs with tensor and sequence parallelism to accelerate the training process.

**Inference details.** During inference, we set the number of diffusion denoising steps to 50 and use a classifier-free guidance scale of 7.5. For quantitative evaluation, we generate 100 videos for each compared method. The text prompts used include both short descriptions (10–20 words) and long

prompts (over 30 words), allowing for a more comprehensive assessment of each model's ability on prompt following.

**Training and inference speed.** For model training, our model was trained on H20 GPUs, with each iteration taking approximately 3 minutes. The full training process consisted of 20,000 iterations, amounting to roughly 40 GPU days in total. Regarding inference speed, using 4 H20 GPUs, generating a video with 50 denoising steps takes approximately 5 minutes. We will include these details in the revised manuscript to provide a clearer picture of the system's computational requirements. Thank you again for your helpful suggestion.

**Evaluation metrics.** To evaluate the performance of video customization, we employ the following metrics to evaluate the identity preservation, text-video alignment, and video generation quality:

- **ID consistency.** We employ Arcface (Deng et al., 2019) to detect and extract the embedding of the reference face and each frames of generation video, and then compute the average cosine similarity between them.

- **Subject similarity.** First, we detect each frame and get the segment result of the subject using YOLOv11 (Khanam & Hussain, 2024), and then compute the similarity of the DINO-v2 (Oquab et al., 2023) feature between the reference and results.

- **Text-video alignment.** We employ CLIP-B and CLIP-L (Radford et al., 2021) to evaluate the alignment between the given text prompt and the corresponding generated videos.

- **Fréchet Video Distance.** FVD first extract the video features through I3D (Carreira & Zisserman, 2017) for the generated videos and target real videos, and then compute the Fréchet Distance between them to evaluate the generation quality and diversity.

- **Temporal consistency.** Following VBench (Huang et al., 2024a), we utilize the CLIP-B (Radford et al., 2021) model to calculate the similarity between each frame and its adjacent frames, as well as the first frame, to assess the temporal consistency of the video.

### A.3 MULTI-MODAL DATA CONSTRUCTION

Our data undergoes a rigorous processing pipeline to ensure high-quality inputs that enhance model performance. Experimental results demonstrate that high-quality data plays a crucial role in tasks such as subject consistency, video editing, and audio-driven video generation. While different tasks may follow their own specific data processing steps, the initial processing stages are common across tasks, with the key differences lying in the subsequent steps. In light of this, this section delves into the detailed methodologies of video data preparation, focusing on the shared preprocessing techniques as well as the task-specific post-processing approaches designed for distinct tasks.

Our data is sourced from diverse channels, and to ensure strict compliance with the principles outlined in the General Data Protection Regulation (GDPR) (Regulation, 2018) framework, we employ data synthesis and privacy-preserving computation techniques to regulate the data collection process. The raw data spans a wide range of domains, primarily encompassing eight major categories: humans, animals, plants, landscapes, vehicles, objects, architecture, and anime. In addition to our self-collected data, we have rigorously curated and processed open-source datasets (e.g., OpenHumanvid (Li et al., 2024b)), which significantly expand the diversity of our data distribution and enhance model performance. Experimental results confirm that the incorporation of high-standard data is crucial for achieving substantial improvements in model performance.

**Data Filtering and Preprocessing.** Given the broad distribution of our dataset, which also includes open-source data, there are significant variations in duration, resolution, and quality among the videos. To address these issues, we implemented a series of preprocessing techniques. Firstly, to prevent transitions within training data, we utilized PySceneDetect (Castellano, 2020) to segment the original videos into single-shot clips. For handling text regions in videos, we employed textbpn-plus-plus (Zhang et al., 2023a) to filter out clips with excessive text and cropped videos containing subtitles, watermarks, and logos. Due to the uneven distribution of video sizes and durations, we performed cropping and alignment, standardizing the short side to either 512 or 720 pixels and limiting video length to 5 seconds (129 frames). Finally, considering that PySceneDetect cannot detect gradual transitions and textbpn-plus-plus (Zhang et al., 2023a) has limited capability in detecting minor text, and to ensure aesthetic quality, motion magnitude, and scene brightness, we used the koala-36M (Wang et al., 2024a) model for further refinement. However, due to differences between

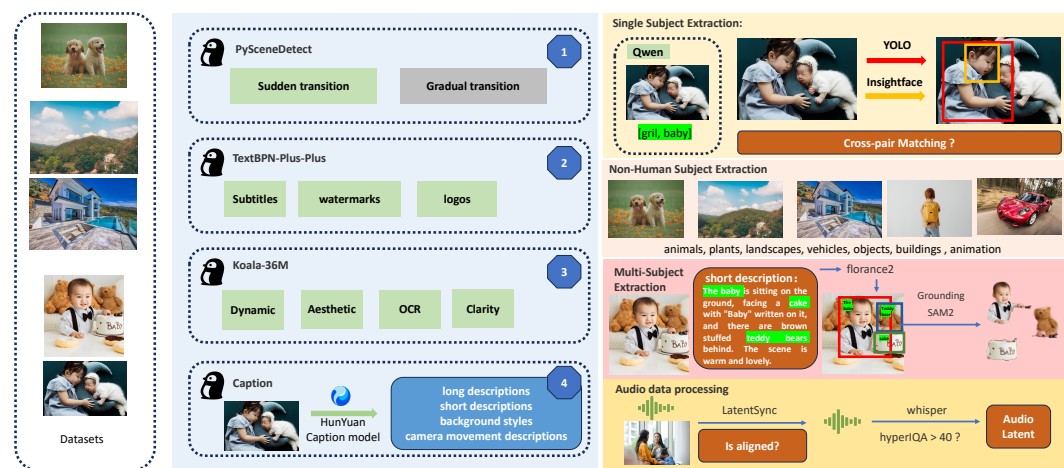

Figure 6: Data Construction Pipeline.

the training data of koala-36M (Wang et al., 2024a) and our dataset, and its lack of fine-grained assessment on aesthetic quality and motion magnitude, we established our own evaluation criteria, determining a koala threshold of 0.06 specific to our dataset for meticulous filtering. Experimental results confirm the importance of our data selection and processing methods in enhancing model performance.

**Subject Extraction.** Single Subject Extraction: To extract the main subject from videos, we first use the Qwen7B (Bai et al., 2023) model to label all subjects in each frame and extract their IDs. Subsequently, we employ a clustering algorithm (e.g., Union-Find) to compute the frequency of each ID's appearance across frames and select the ID with the highest occurrence as the target subject. Multiple IDs can be chosen if necessary; however, if all IDs appear fewer than a predefined threshold (e.g., 50 frames), the video is discarded. Next, we use YOLO11X (Khanam & Hussain, 2024) for human body segmentation to obtain bounding boxes and InsightFace (Ren et al., 2023) to detect face positions and generate face bounding boxes. If the proportion of the face bounding box within the human body bounding box is less than 0.5, the detection result from YOLO11X is considered erroneous, and the corresponding bounding box is discarded.

Non-Human Subject Extraction: For non-human subjects, we utilize QwenVL (Bai et al., 2025b) to extract subject keywords from the video and employ GroundingSAM2 (Ravi et al., 2024; Liu et al., 2023b; Ren et al., 2024a;b; Jiang et al., 2024b) to generate masks and bounding boxes based on these keywords. If the size of a bounding box is less than 0.3 times the dimensions of the source video, it is discarded. To ensure balanced category distribution in the training data, we use QwenVL to classify the main subject into one of eight predefined categories: animals, plants, landscapes, vehicles, objects, architecture, and anime. We then apply balanced sampling across these categories to achieve an equitable distribution.

Multi-Subject Extraction: For multi-subject scenarios, we use QwenVL to filter videos from single-person datasets that involve interactions between humans and objects. Since we need to align the subject keywords in video captions with those in images, directly using QwenVL to re-extract subject keywords may lead to misalignment with the keywords in the video prompt. Therefore, we employ Florence2 (Xiao et al., 2024) to extract bounding boxes for all subjects mentioned in the video captions. Subsequently, GroundingSAM2 is used to perform subject extraction on these bounding box regions. We then apply clustering to remove frames that do not contain all subjects. To address issues related to hard-copying, we use the first 5 seconds of the video for model training and the subsequent 15 seconds for subject segmentation.

**Video Resolution Standardization.** We first compute a union bounding box based on all the bounding boxes of the main subjects and ensure that the cropped region contains at least 70% of the area of the union bounding box. To enable the model to support multi-resolution outputs, we define several aspect ratios, including 1:1, 3:4, and 9:16.

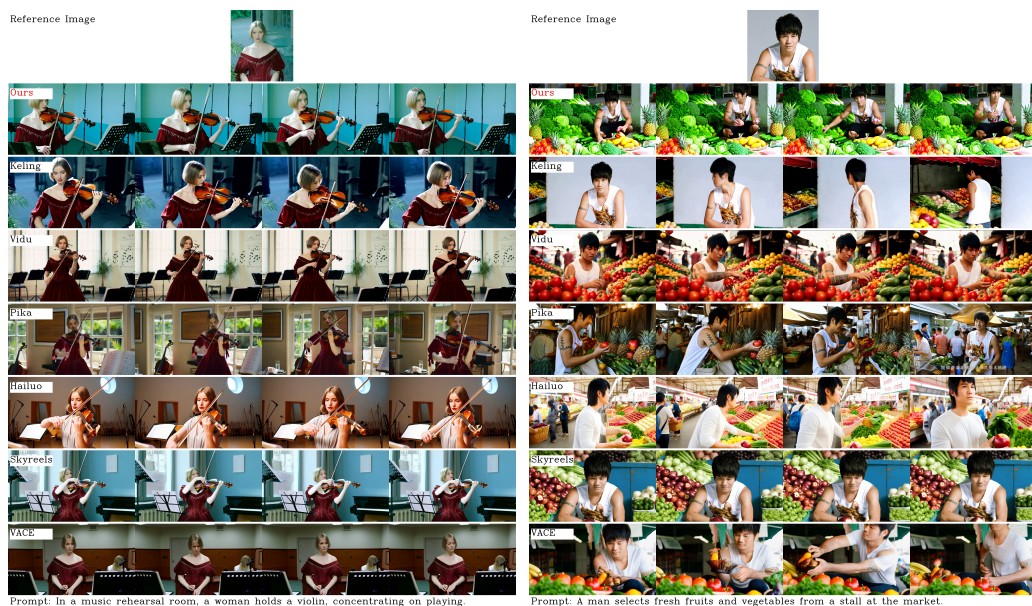

Figure 7: Comparison on human-centered video customization.

**Video Annotation.** We employ a structured video annotation model to label the videos. This model provides detailed descriptive information, including long descriptions, short descriptions, background styles, and camera movement descriptions of the videos. During the training process, these structured annotations are utilized to enhance the video captions, thereby improving the robustness and performance of the model.

**Mask Data Augmentation.** During video editing, directly using the extracted subject masks for training can lead to overfitting when replacing objects of different types or shapes. For instance, replacing a doll without ears with one that has ears might result in the generated video still showing the doll without ears, which is not the desired outcome. Therefore, during the training process, we apply techniques such as mask dilation or converting masks to bounding boxes to soften the mask boundaries. These methods help achieve more realistic and expected editing results in the final video. By employing these augmentation strategies, we aim to mitigate overfitting issues and ensure that the edited videos meet our expectations more closely. This approach enhances the flexibility and applicability of the model across various object types and shapes.

**Audio data processing.** First, we utilize LatentSync (Li et al., 2024a) to evaluate the synchronization between audio and video in the clips. Specifically, we discard videos with a synchronization confidence score below 3 and adjust the audio-video offset to zero. At the same time, we compute the hyperIQA quality score and remove any videos scoring below 40 to ensure high-quality data. Finally, we employ Whisper (Radford et al., 2023) to extract audio features, which will be used as input for subsequent model training.

## A.4 MORE SINGLE-SUBJECT COMPARISON RESULTS

In this section, we present additional comparisons on single-subject customization, which includes both human and object customization tasks. We compare OmniCustom with several state-of-the-art methods, including Pika (Pika, 2025), Vidu (Vidu, 2025), Keling (Keling, 2025), VACE 1.3B (Jiang et al., 2025), and Skyreels A2 (Fei et al., 2025). Additionally, since Hailuo only supports human customization, we further include Hailuo in the human-centered video customization comparison. The results of the human-centered comparisons are shown in Fig. 7. As illustrated, VACE, SkyReels-A2, Vidu, Hailuo, and Pika all exhibit poor identity preservation. Among these, only Keling demonstrates relatively better identity consistency. However, Keling suffers from a copy-paste artifact: for example, in the rightmost case, the man appears with a gray background copied directly from the input image, which does not blend naturally with the background scene. For object-centered customization, the

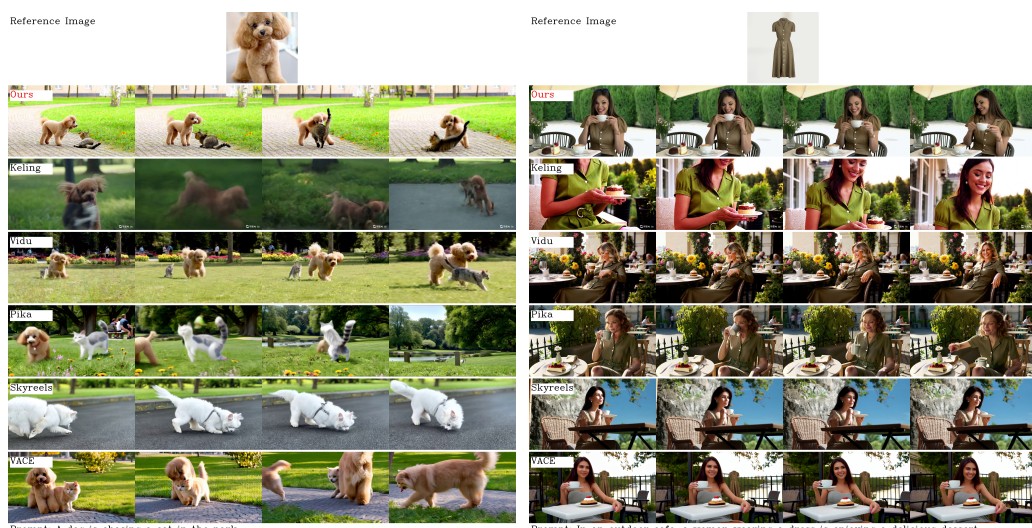

Figure 8: Comparison on object-centered video customization.

| Metric | Face-sim ↑ | DINO-sim ↑ | CLIP-B ↑ | CLIP-L ↑ | FVD ↓ | Temporal ↑ |
|---|---|---|---|---|---|---|
| VACE | 0.433 | 0.598 | 0.335 | 0.280 | 1171 | 0.966 |
| SkyReels-A2 | 0.554 | 0.619 | 0.332 | 0.276 | 1379 | 0.943 |
| Keling-1.6 | 0.534 | 0.554 | 0.330 | 0.280 | 1049 | 0.934 |
| Vidu-2.0 | 0.532 | 0.588 | **0.336** | **0.282** | 1083 | **0.970** |
| Pika | 0.546 | 0.548 | 0.310 | 0.263 | 980 | 0.942 |
| OmniCustom | **0.630** | **0.622** | 0.331 | 0.274 | **971** | 0.962 |

results are presented in Fig. 8. Keling tends to generate blurry outputs, while Skyreels A2 fails to accurately capture the target subject, producing incorrect colors for both the dog and the dress. VACE also struggles with subject consistency, failing to generate the green dress and merging the dog and cat into a single, unnatural subject. Vidu and Pika achieve relatively better subject consistency, but still lose some fine details of the target object (e.g., the buttons on the dress). In contrast, our method achieves the best subject-consistent generation in both human-centered and object centered video customization, preserving both the global visual appearance and local details, while maintaining high overall generation quality.

## A.5 QUANTITATIVE MULTI-SUBJECT COMPARISON RESULTS

Tab. A.5 summarizes the quantitative performance of different video customization methods on the multi-subject benchmark. OmniCustom consistently achieves the highest identity preservation scores, as reflected in both Face-sim (0.630) and DINO-sim (0.622), surpassing all baselines by a notable margin. While Vidu-2.0 Vidu (2025) demonstrates competitive results in prompt adherence (CLIP-B and CLIP-L), OmniCustom delivers comparable CLIP scores, indicating strong ability to follow textual instructions without compromising subject consistency. For generation quality and temporal stability, OmniCustom attains the lowest FVD score (971), outperforming methods such as Pika Pika (2025) (980) and Vidu-2.0 (1083), and maintains robust temporal consistency (0.962), second only to Vidu-2.0 (0.970). Notably, baseline methods such as VACE Jiang et al. (2025) and SkyReels-A2 Fei et al. (2025) obtain lower identity similarity scores and higher FVD values, suggesting inferior ability in preserving subject characteristics and global video quality. Overall, OmniCustom demonstrates a superior balance of identity consistency, video quality, prompt alignment, and temporal coherence, highlighting its effectiveness for challenging multi-subject video customization tasks.

## A.6 MORE RESULTS ON AUDIO-DRIVEN VIDEO CUSTOMIZATION

**More qualitative results.** Previous audio-driven human animation methods input a human image and an audio, where the human posture, attire, and environment remain consistent with the given image

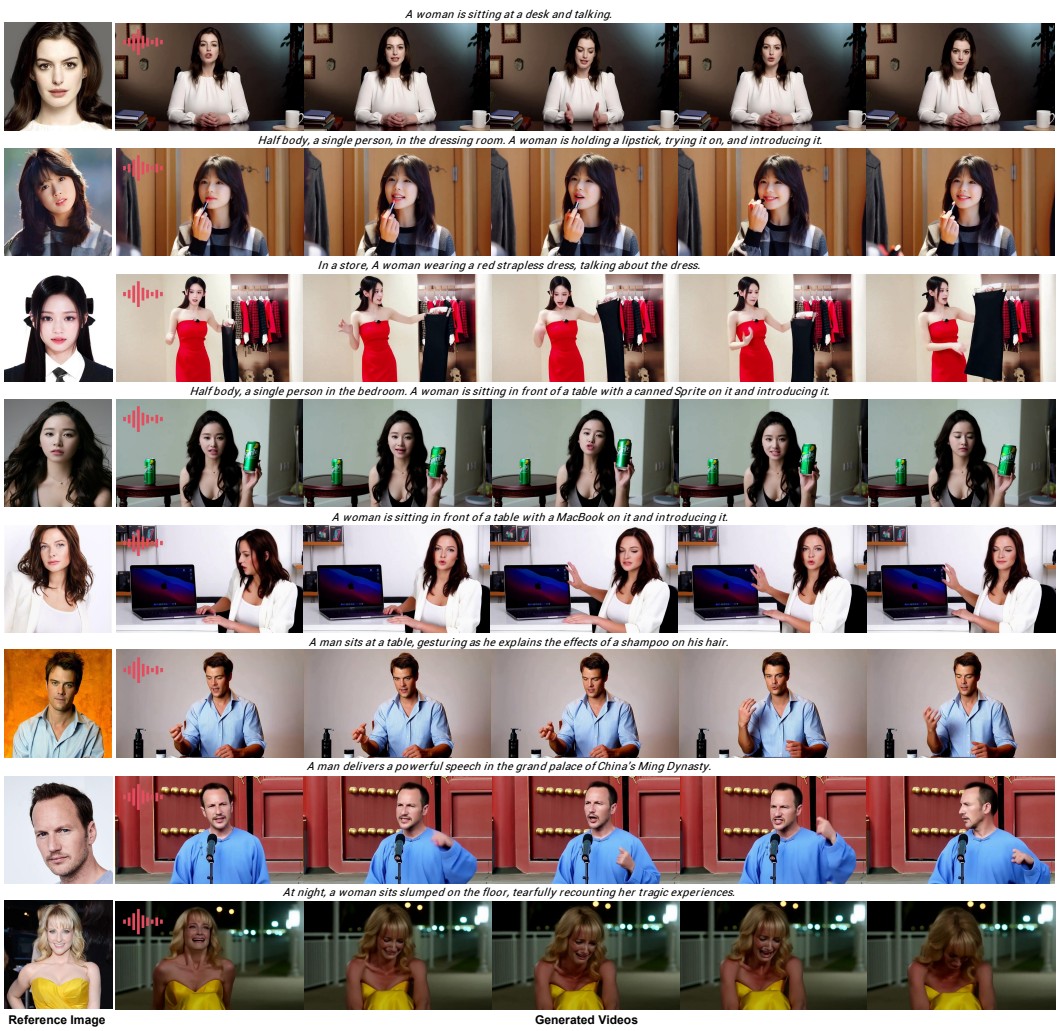

Figure 9: The results of our OmniCustom in Audio-driven customization, where we can generate videos in different scenes and postures specified by the text prompt, while keeping the identity well.

and cannot generate videos in other gestures and environments, which may restrict their application. In comparison, our OmniCustom enables audio-driven human customization, where the character speaks the corresponding audio in a text-described scene and posture, allowing for more flexible and controllable audio-driven human animation. We show more audio-driven video customization results in Fig. 9. OmniCustom produces videos that closely align with the given prompts while preserving character identities. It demonstrates effective interaction with other subjects (rows 3 & 4) or humans (rows 5 & 6), which can significantly enhance its application in live streaming and advertising. Additionally, it can generate videos featuring diverse scenes and postures, such as those set in the Ming Dynasty (row 7), where characters are automatically dressed in period-appropriate attire without explicit prompts, and row 8 showcases a woman with vivid and realistic expressions distinct from the input image. This demonstrates OmniCustom's robust world modeling and generalization capabilities. In summary, our audio-driven OmniCustom can generate videos across various scenes and postures specified by text prompts with high diversity, while keeping the identity well.

**Comparison on Lip-sync Performance.** To further assess the effectiveness of our audio-driven customization approach, we conduct a comparative evaluation against existing state-of-the-art audio-driven portrait animation methods, including EchoMimic (Chen et al., 2025b), EchoMimic-V2 (Meng et al., 2025), and Hallo-3 (Cui et al., 2025). The comparison is performed using the widely adopted

Table 3: Lip-sync accuracy (Sync-C ↑) comparison with existing methods.

| Models | OmniCustom | EchoMimic | EchoMimic-V2 | Hallo-3 |
|---|---|---|---|---|
| Sync-C | 4.41 | 3.41 | 4.11 | **4.57** |

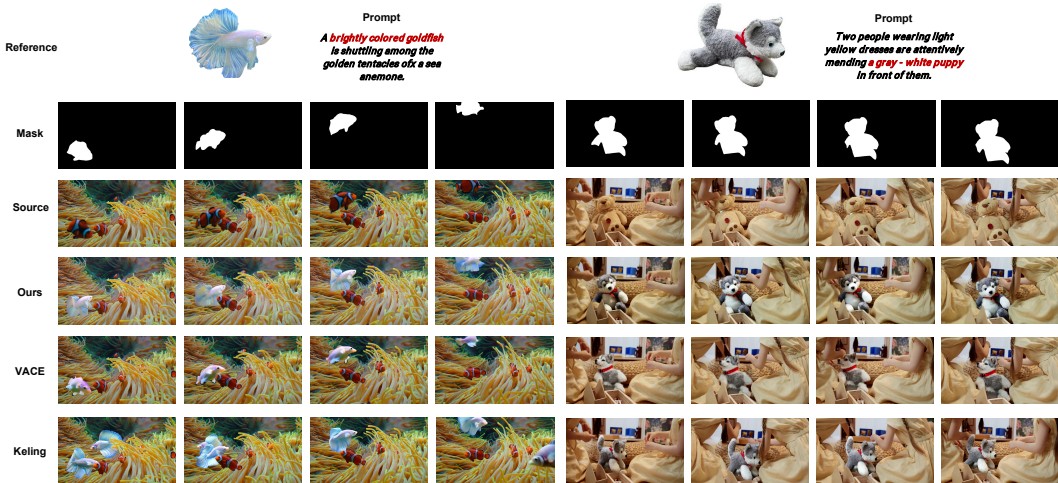

Figure 10: The results of our OmniCustom in Video-driven video customization, where we can edit anything in the source video with a given mask video, while generating video vividly.

Sync-C metric (Chung & Zisserman, 2016), which measures lip-sync accuracy. It is important to point out that our audio-driven customization task is inherently more challenging than traditional audio-driven portrait animation, as it involves generating not only accurate lip synchronization but also modeling environmental and subject interactions. Therefore, a direct comparison is meaningful primarily in terms of lip-sync accuracy. As presented in Table 3, our method achieves competitive Sync-C scores, demonstrating its strong capability in audio-synchronized video generation.

### A.7 MORE RESULTS ON VIDEO-DRIVEN VIDEO CUSTOMIZATION

Leveraging its strong subject consistency, OmniCustom also supports video-driven video editing, enabling a broad range of application scenarios. We compare OmniCustom with VACE (Jiang et al., 2025) and Keling (Keling, 2025) on the task of video subject replacement, where a source video, object masks indicating regions to be replaced, and a target subject image are provided as inputs. The results are presented in Fig. 10. VACE suffers from boundary artifacts due to strict adherence to the input masks, resulting in unnatural subject shapes and disrupted motion continuity. Keling, in contrast, exhibits a copy-paste effect, where subjects are directly overlaid onto the video, leading to poor integration with the background. In comparison, OmniCustom effectively avoids boundary artifacts, achieves seamless integration with the video background, and maintains strong identity preservation—demonstrating its superior performance in video editing tasks.

### A.8 ROBUSTNESS TO DIVERSE REAL-WORLD SCENARIOS

To address concerns regarding the diversity of scenario testing and further validate the robustness of our model, we conducted additional experiments on more challenging real-world conditions, including low-light environments and rapid motion involving multiple subjects. As shown in Table A.8, our model maintains strong performance across all tested scenarios. In low-light and rapid motion settings, there is a slight decline in certain metrics, such as Face-Sim and FVD, which is expected due to increased scene complexity. Nevertheless, OmniCustom consistently produces plausible and coherent video content, demonstrating its ability to generalize beyond standard conditions. The model's temporal consistency remains high even under adverse settings, and identity preservation and

| Models | Face-Sim ↑ | DINO-Sim ↑ | CLIP-B-T ↑ | CLIP-L-T ↑ | FVD ↓ | Temp-Consis ↑ |
|---|---|---|---|---|---|---|
| OmniCustom | **0.624** | **0.607** | **0.309** | **0.261** | **1305** | 0.961 |
| OmniCustom (Low-light) | 0.605 | 0.591 | 0.304 | **0.261** | 1587 | **0.963** |
| OmniCustom (rapid motion) | 0.598 | 0.584 | 0.303 | 0.256 | 1492 | 0.926 |

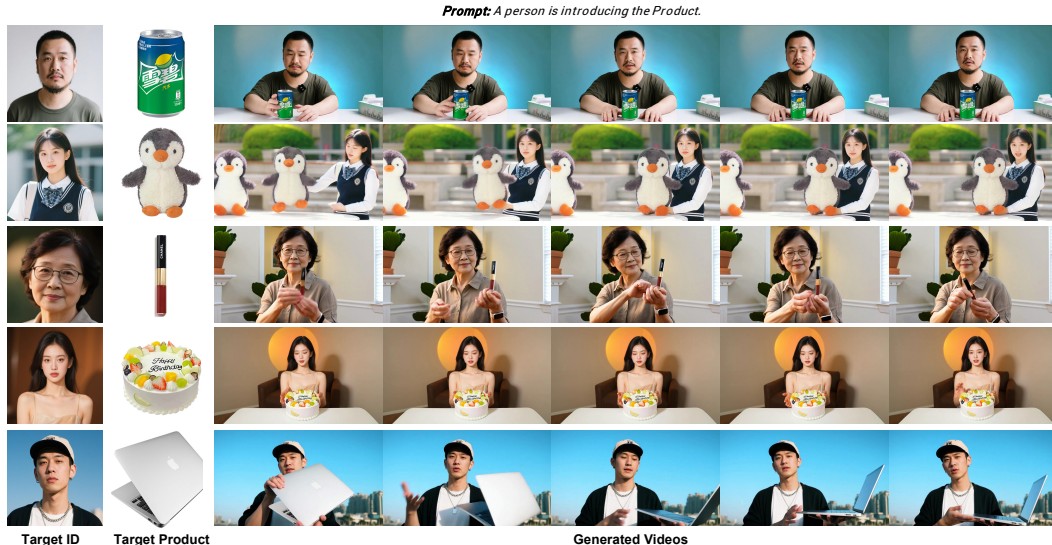

Figure 11: The results of our OmniCustom in virtual human advertisement, where OmniCustom can generate advertisement videos with good interaction between the human and products.

prompt adherence are only minimally affected. These results highlight the robustness and adaptability of our approach, underscoring its potential applicability in diverse and complex real-world scenarios.

### A.9 MORE APPLICATIONS

**Virtual Human Advertisement.** Leveraging our multi-subject customization capability, Omni-Custom enables applications that previous methods cannot achieve. A significant application is in virtual human advertising, where OmniCustom takes a human image and a product image as inputs to generate a corresponding advertisement video. The results, shown in Fig. 11, demonstrate that OmniCustom effectively maintains the identity of the human while preserving the details of the target product, including the text on it. Furthermore, the interaction between the human and the product appears natural, and the video adheres closely to the given prompt, highlighting the substantial potential of OmniCustom in generating advertisement videos.

**Audio-driven virtual try-on.** Utilizing its multi-subject customization capability, OmniCustom also supports audio-driven multi-subject video customization, offering a wide range of applications. In the main paper, we demonstrated OmniCustom's capabilities in virtual human advertising. Here, we further explore its generation ability in virtual try-on, driven by both text prompts and audios. The results, shown in Fig. 12, illustrate the integration of virtual try-on with audio-driven video generation. The generated videos effectively preserve the target identities while naturally the specified attire and synchronizing vividly with the given audio. This highlights OmniCustom's robust capability in multi-modal video customization.

### A.10 LIMITATIONS AND SOCIETAL IMPACTS

**Limitations.** Current video generation models (Kong et al., 2024; Wang et al., 2025) still face challenges in accurately modeling the real world, particularly when it comes to capturing complex physical rules and interactions. Therefore, when generating customized videos involving multiple subjects engaged in intricate interactions, OmniCustom sometimes struggles to faithfully represent the relationships between subjects. As a result, artifacts may appear in both the visual appearance

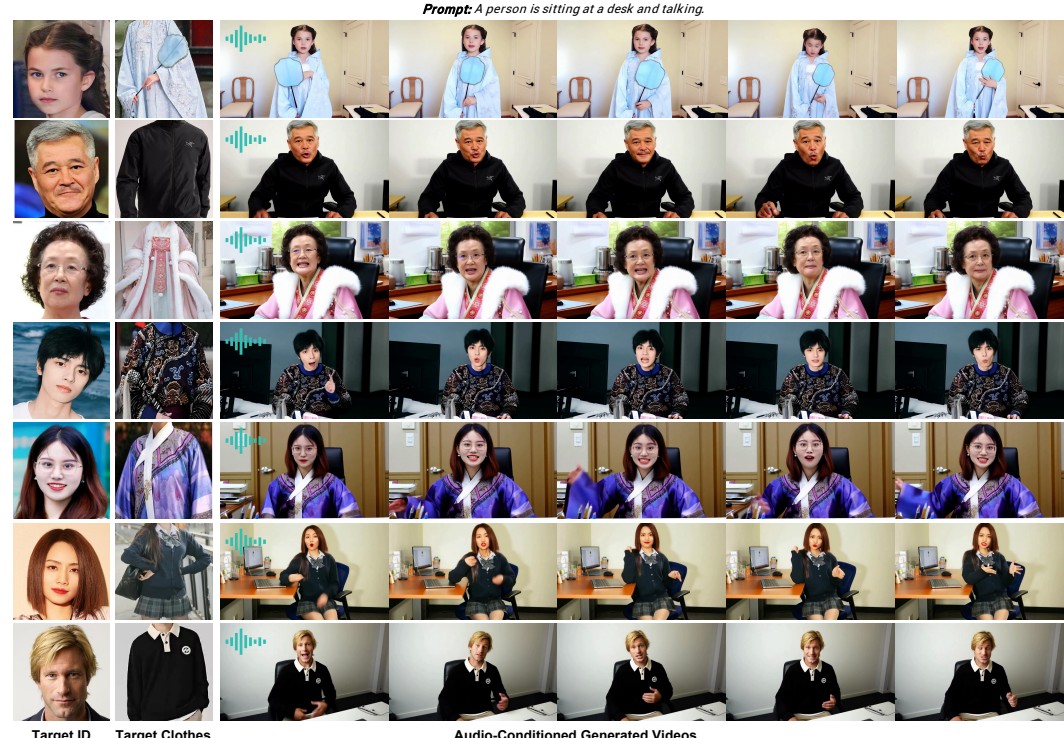

Figure 12: The results of our OmniCustom in audio-driven multi-subject customization, where we can generate humans in different clothes, while speaking the given audio vividly.

and motion of the generated subjects. For example, the model may fail to maintain correct spatial relationships, leading to unnatural overlaps or collisions, or it may generate unrealistic motions that do not adhere to the expected physical dynamics. These limitations highlight the need for further research to improve the ability of video generation models to understand and reproduce complex multi-subject interactions in a realistic manner.

**Societal impacts.** The development of OmniCustom, a controllable and multi-modal video generation model, has the potential to greatly benefit society by lowering the barriers to high-quality, personalized video creation for entertainment, education, advertising, and more. Its ability to generate subject-consistent videos from diverse inputs can empower both individuals and organizations to express creativity and communicate ideas more effectively. However, this technology also raises concerns about potential misuse, such as the creation of deepfakes or unauthorized use of personal likenesses. It is therefore important to promote responsible use and develop safeguards to ensure that such advancements contribute positively to society.

