# OpenReview forum: "OmniCustom: A Multimodal-Driven Architecture for Customized Video Generation"
_ICLR.cc/2026/Conference — ICLR 2026 Conference Withdrawn Submission_

### Official Review · Reviewer_K8PH · 2025-10-15

**Soundness:** 3
**Presentation:** 2
**Contribution:** 2
**Rating:** 4
**Confidence:** 4

**Summary:**

This paper presents OmniCustom, a multi-modal customized video generation model, designed to address two key limitations of existing methods: poor subject identity consistency and restricted input modalities. Built on the HunyuanVideo framework, OmniCustom supports flexible video generation conditioned on text, images, audio, and video inputs and maintains high subject consistency across single- and multi-subject scenarios. Key technical components include: An identity-enhanced text-image conditioning module (integrating LLaVA for multi-modal understanding and temporal concatenation to reinforce identity features). Identity-disentangled injection modules: AudioNet: Aligns audio features with video latents via spatial cross-attention for audio-driven generation.
Video injection module: Projects conditional video into the latent space and aligns features efficiently for video-driven editing.
A rigorously constructed multi-modal dataset (filtered for quality, with subject segmentation and audio-video synchronization).
Experimental results show OmniCustom outperforms state-of-the-art methods (e.g., Vidu, Pika, VACE) in identity consistency, realism, and text-video alignment. It also demonstrates robustness in downstream applications like virtual human advertising, virtual try-ons, and video editing.

**Strengths:**

Strong multi-modal support: Unlike existing single-modality (image-driven) methods, it natively handles text, image, audio, and video inputs, enabling flexible customization (e.g., audio-driven speech animation, video-driven subject replacement).

Superior identity consistency: The identity-enhanced module and disentangled injection design preserve subject details (both human and non-human) across frames, outperforming baselines in metrics like Face-Sim and DINO-Sim.

Efficient computation: The video injection module uses feature-alignment addition (instead of adapter-based conditioning or temporal concatenation) to avoid extra computational overhead during inference.

**Weaknesses:**

1. No video demo is presented, the performance is hard to validate.

2. Using rope to differentiate different identities sounds strange.

3. The overall frameworks seems identical with existing works.

**Questions:**

Please refer to the weaknesses part.

---

### Official Review · Reviewer_Yd2L · 2025-10-20

**Soundness:** 3
**Presentation:** 3
**Contribution:** 2
**Rating:** 4
**Confidence:** 4

**Summary:**

The authors propose OmniCustom, a multi-modal driven video customization framework. The framework supports multi-modal conditions for generating customized videos, including images, audio, masks, video, and text. Specifically, the framework consists of an identity-enhanced text–image condition module, an identity-disentangled AudioNet, and an identity-disentangled video injection module. The training datasets include Koala-36M and a private dataset. The base video generation model used is HunyuanVideo.

**Strengths:**

1. The paper is clearly structured and easy to follow.
2. The input conditions supported are diverse and multi-modal.
3. The proposed method appears straightforward and effective.

**Weaknesses:**

1. No website or demo is provided in the supplementary material, making it difficult to assess the quality of the generated videos.
2. The novelty of the method seems limited, as the framework appears to directly build upon the HunyuanVideo base model with several additional encoders/tokenizers.
3. There are no quantitative comparisons with existing models for multi-condition video generation.

**Questions:**

Could the authors provide the parameter counts and generation times for OmniCustom and each of the compared models? Without this information, it is difficult to assess the fairness of the comparisons. For example, comparing OmniCustom with VACE 1.3B is not equivalent to comparing it with VACE 14B, as the difference in model scale may significantly affect performance and efficiency.

---

### Official Review · Reviewer_PTwo · 2025-10-31

**Soundness:** 2
**Presentation:** 2
**Contribution:** 1
**Rating:** 2
**Confidence:** 3

**Summary:**

This paper introduces OmniCustom, a framework that enables video customization using multi-modal inputs, including image, audio, mask, and video.

**Strengths:**

The paper explores a range of input modalities, demonstrating the capability to generate videos customized with various types of input data.

**Weaknesses:**

- The claim of "omni-modal" support is somewhat overstated, as the different input modalities are handled separately rather than via a unified framework. Furthermore, previous works have already addressed similar types of inputs and tasks.
- The introduced task closely resembles digital human video generation, which typically utilizes audio, facial images, and text prompts. The use of an identity-disentangled AudioNet for audio injection is similar to existing approaches, and audio is treated as optional. In addition, most tasks presented in the paper could be achieved through existing digital human video generation methods. For instance, multi-subject customization via continual tuning for image concept learning is a standard technique in the literature.
- The evaluation is limited, with comparisons restricted to video foundation models. The paper should include comparisons with established methods in video personalization and talking face video generation to better situate its contributions.

**Questions:**

n/a

---

### Note · Authors · 2025-11-14

I have read and agree with the venue's withdrawal policy on behalf of myself and my co-authors.